# Implementation of a smoking cessation intervention for people with severe mental illness in ambulatory mental healthcare (KISMET): A process evaluation

Müge H. Küçükaksu[1]*, Lola Jansen[1], Trynke Hoekstra[1], Sanne Helmig[1], Marcel C. Adriaanse[1], Berno van Meijel[2,3,4]

**1** Department of Health Sciences and Amsterdam Public Health research institute, Vrije Universiteit, Amsterdam, Netherlands, **2** Department of Psychiatry, Amsterdam UMC and Amsterdam Public Health research institute, Amsterdam, Netherlands, **3** Inholland University of Applied Sciences, Centre of Expertise Prevention in Health and Social Care, Faculty of Health, Sports and Social Work, Amsterdam, The Netherland, **4** Parnassia Psychiatric Institute, The Hague, The Netherlands

\* m.kucukaksu@vu.nl

## Abstract

### Background

Recently, a cluster-randomised controlled trial was conducted within Dutch ambulatory mental healthcare to assess the effectiveness of a one-year smoking cessation intervention consisting of group sessions, peer support and pharmacological treatment (KISMET). This article presents its process evaluation, exploring the perceptions of patients and mental healthcare professionals (MHPs) regarding the implementation of KISMET.

### Methods

We conducted a mixed methods study, including 26 semi-structured interviews (10 MHPs and 16 patients) following the RE-AIM framework. Qualitative data was thematically analysed with MAXQDA software. We collected additional data on inclusion, drop-out and assessed treatment adherence during on-site observations.

### Results

Recruitment and subsequent retention of patients in the RCT was found to be challenging (58% drop-out at one-year follow-up). MHPs suggested more motivational enhancement techniques to aid recruitment and reduce study attrition. The intervention components were generally assessed positively. Patients experienced the group and peer support sessions as supportive and beneficial. Inconsistent group attendance was demotivating and disrupted implementation. Pharmacological treatment was found to be helpful, although MHPs mentioned the treating psychiatrist

**Data availability statement:** Qualitative data of this study are not publicly available due to restrictions imposed by the Vrije Universiteit Amsterdam, due to their sensitive content and patient confidentiality. However, data can be made available by the corresponding author on reasonable request and by contacting dr. Barbara Groot-Sluijsmans (secretariaat.agw.beta@vu.nl), the qualitative research and data expert of our department. To ensure long-term data storage and availability, we store the transcripts and analysis files on a password-protected drive to which only the research team and the non-author have access to.

**Funding:** This research is funded by ZorgOnderzoek Nederland Medische Wetenschappen (ZonMw) and Stichting tot Steun Vereeniging tot Christelijke Verzorging van Krankzinnigen en Zenuwlijders (grant number 258). Funds were received by MCA.

**Competing interests:** The authors have declared that no competing interests exist.

or clinical nurse specialist need to facilitate better to ensure medication supply. The handbook offered MHPs excellent guidance and significantly contributed to treatment fidelity. Surprisingly, the carbon monoxide monitoring (originally included in the study protocol for research purposes) was appraised as a motivational tool. Finally, shortage of staff, limited collaboration and high turnover were barriers for the delivery of the intervention. Implementation heavily depends on the quality of the collaboration between MHPs and patients, as well as the infrastructure created by the organisation.

## Conclusion

This study demonstrates the benefits, unique demands and challenges of a smoking cessation intervention for people with SMI. Results could guide and improve the implementation of smoking cessation interventions in mental healthcare settings. Fostering a culture of connectivity through team development, along with the provision of supportive and communicative supervision is critical for the effective recruitment and retention in smoking cessation studies in psychiatric care.

## Introduction

Smoking poses an immediate health risk and accounts for various diseases such as cardiovascular diseases, cancers and respiratory diseases. These risks are relatively higher among people with severe mental illness (SMI) due to heavier smoking and more years of smoking compared to people without SMI [1]. Smoking is also two to three times more prevalent among people with SMI [1]. This leads to larger health disparities and high premature mortality rates [2–4]. It is known that smoking is an exacerbating factor for psychiatric symptoms in schizophrenia, bipolar disorder and depression, additionally showing the urgency to integrate smoking cessation support in mental healthcare [5–7].

Unfortunately, the delivery of smoking cessation interventions within mental healthcare settings is still limited. The implementation of these interventions is often hindered by factors such as lack of financial resources, shortage in personnel and low prioritisation by mental healthcare organisations or individual mental healthcare professionals (MHPs). Further, there generally exists a lack of knowledge about and experience with treating tobacco addiction [8,9]. Despite these barriers, several mental healthcare organisations in the Netherlands, among other countries, have attempted to allocate more attention and resources to tackle the high prevalence of tobacco addiction among people with SMI. This aligns with the Dutch national agreement to reduce tobacco use [10]. Mental health care organisations that have signed this agreement set the goal to be smoke-free institutions by 2025. This includes buildings and terrains to be smoke-free, MHPs to not smoke while at work, and MHPs as well as patients to have more opportunities to receive support when wishing to quit smoking.

The effectiveness of cognitive behavioural therapy (CBT) and motivational interviewing (MI) techniques, combined with pharmacological support, such as nicotine replacement therapy (NRT) for smoking cessation in people with severe mental illness has been studied previously, although not in combination with additional peer support [11–17]. However, the effectiveness of an intervention is not only based on its therapeutic quality, but also depends on its practical implementation. An optimal implementation within a mental healthcare organisation requires an interplay of multiple factors on different levels, such as prioritisation, allocation of financial resources to implement the intervention, sufficient available and qualified professionals, their affinity with smoking cessation, and effective teamwork. More evidence is needed on how the implementation of smoking cessation interventions in clinical practice can be promoted and optimised to ensure the best possible treatment effect.

Between September 2022 and June 2024, we implemented a comprehensive intervention for smoking cessation (the KISMET intervention) treated by Flexible Assertive Community Treatment (FACT) teams in the Netherlands [18]. For one year, FACT teams delivered a combined smoking cessation programme consisting of group sessions, peer support and (optional) pharmacological treatment to patients with a desire to quit smoking. This article presents the perceptions of patients and MHPs of the implementation of the KISMET intervention.

## Methods

### Study design

This process evaluation entailed mixed-methods design, including semi-structured interviews as part of the pragmatic cluster-randomized controlled trial (RCT). Data triangulation was used by combining three data sources: interviews with MHPs and patients assigned to the KISMET intervention, on-site observations of group sessions, and descriptive data from the RCT.

### KISMET intervention

In total, 11 FACT teams offered the one-year KISMET intervention to their patients. Mental healthcare nurses, social workers, psychologists, psychiatrists, and experts-by-experience were involved in the implementation of the intervention. Ten FACT teams were in the control condition of the trial and provided treatment as usual (TAU) [19].

The KISMET intervention consists of three components: 1. group sessions based on cognitive behavioural therapy (CBT) and motivational interviewing (MI) techniques; 2. peer support group meetings led by an expert-by-experience; 3. (optional) pharmacological treatment. If participants decided to use pharmacotherapy, they could choose between nicotine replacement therapy (NRT) and standard medication for smoking cessation (i.e., Nortriptyline, Bupropion, Cytisine) according to national guidelines [20,21]. Before the start of the intervention, MHPs received a one-day training for the effective implementation of the intervention. In the first three months, MHPs offered intensive support through weekly group sessions and peer support meetings. After this initial phase, monthly booster sessions took place recapping previous sessions and discussing new challenges and successes (maintenance phase). The overall goal was to prepare for a quit attempt or to gradually reduce the number of cigarettes per day, depending on patients' personal goals. The intervention was systematically developed using a modified Delphi procedure [22]. The content of the sessions is protocolised, and written out in a handbook by the authors [22]. An overview of the sessions' content can be found in Appendix I in S1 File. The methods and design of the KISMET study have been described previously [18].

### MHPs and patient selection and recruitment

In the period between November 2023 and February 2024, we interviewed MHPs who carried out the KISMET intervention within their FACT teams. We also conducted interviews with patients to understand their perceptions of the implementation of the intervention, including facilitators and barriers. For the selection of FACT team members and patients for the

interviews we applied purposive sampling methods [23]. To be eligible for an interview, patients and MHPs had to be part of FACT teams allocated to the intervention arm of the RCT, with patients still being enrolled in the study and not having dropped out. MHPs had to be actively involved in providing the group or peer support sessions. We intended to select a sample that is as heterogeneous as possible to capture a wide range of perspectives and experiences.

Variation within the sample of MHPs referred to occupational background, age, gender, years of work experience in mental health care and reported implementation success. The latter was defined as achieving successful recruitment (>7 patients included) and completing 20 sessions as protocolised. Variation within the patient sample was achieved by considering the primary mental health diagnosis, age, gender, attendance at sessions, and smoking status after 12 months. Recruitment took place between 15th November 2023 and 8th February 2024. Out of 13 MHPs who were approached, 10 MHPs eventually were interviewed. Three did not respond or were not willing to be interviewed. Out of 22 patients who were approached, 16 patients responded and agreed to an interview. All participants received an information letter about the study and gave their written consent by signing an informed consent.

Individual interviews with patients were held in their home environment or at the mental healthcare institution. The duration of the interviews ranged from 28 minutes to 66 minutes, with an average duration of 47 minutes. Individual interviews with MHPs were all held online using Zoom [24] and lasted on average 56 minutes, with a minimum of 34 minutes and a maximum of 78 minutes.

## Data collection

**Descriptive data.** We included data on patients' attendance at group sessions as well as demographic information and primary DSM-5 diagnosis that were collected during the trial. For professionals, the total number of sessions they delivered and drop-outs among MHPs were registered during the RCT. These data were used as background information for the interviews conducted.

**Interviews and RE-AIM theoretical framework.** Data were collected via semi-structured interviews applying the RE-AIM framework for the evaluation of the implementation. This framework was also used for the design of the interview guide and topics [25,26]. The RE-AIM framework proposes a critical assessment of five dimensions regarding implementation: reach, efficacy, adoption, implementation and maintenance. These dimensions will be briefly explained here.

*Reach.* This dimension assesses to what extent the target population is reached and willing to participate. We evaluated how FACT teams approached the recruitment of patients and to what extent the intervention meets the FACT population's needs. In line with this, we explored opportunities for improving access and appropriateness of the intervention to meet the needs of people with SMI.

*Efficacy.* In the context of this process evaluation, efficacy refers to the effect of the KISMET intervention on smoking (cessation), quit attempts, and awareness of one's own smoking behaviour. We systematically questioned the three components of the intervention and their perceived effectiveness. Additionally, we investigated how carbon monoxide (CO) monitoring, which was part of regular study assessments, was utilised and perceived by patients and MHPs.

*Adoption.* This dimension refers to the settings and temporal circumstances of mental health organisations that were willing to implement the intervention. In our research, we explored how policies and culture regarding smoking within institutions might have created the conditions for FACT teams to be motivated to adopt the intervention. Moreover, we investigated how MHPs experienced the preparatory training, as well as their motivation and willingness to deliver the intervention.

*Implementation.* This dimension refers to MHPs' fidelity to the original intervention protocol and identifies which adjustments to the protocol MHPs made. In addition to the interviews, we used the observations that the researchers (MK and LJ) made during the group sessions they joined to assess fidelity. This dimension also encompasses facilitators and challenges experienced by MHPs while implementing the intervention.

 

*Maintenance.* This dimension indicates the extent to which mental health care organisations or FACT teams intend to incorporate the intervention after the KISMET trial has ended.

A more detailed overview of the topics and corresponding interview questions can be found in Appendix II in <u>S1 File</u>.

**On-site observations.** Complementary to the interviews with MHPs and patients we joined one group session and one peer support group meeting in nine out of 11 intervention teams. One FACT team decided against this as they expected a possibly negative impact on the group dynamics and the comfort of their patients. We collected information about the interaction between MHPs and patients, group dynamics, and proactivity and involvement of patients. We assessed fidelity to the KISMET intervention protocol regarding the extent to which the session adhered to the structure and content by scoring them 'completely/ partly/ not at all implemented'. The most frequent and notable observations from our on-site observations were summarised. Observations were registered with a pre-defined form (Appendix III in <u>S1 File</u>).

## Data analysis

Descriptive data included age, gender, primary diagnosis, attendance at sessions and sessions given by MHPs. These descriptive data were analysed using SPSS version 29 software.

For qualitative analysis, the interviews were transcribed verbatim. Deductive and inductive thematic analysis was applied following six phases of analysis [27]: 1) Familiarisation with the dataset by listening to all interviews, summarising them and reading the transcripts thoroughly. The transcripts were then read and coded by two authors independently (MK and LJ) using MAXQDA software [28]. 2) Generation of initial codes; both researchers (MK and LJ) generated an initial code structure. 3) Constructing themes; 4) reviewing themes and validating them; 5) defining and naming themes; and 6) producing the report with main findings. Initially, a deductive method was applied to generate a coding framework based on the RE-AIM framework and the interview guide. During data analysis, an inductive method was additionally applied to generate new codes for new emerging relevant themes. The process of data analysis and (interim) findings were periodically discussed with the whole research team until a consensus about the main themes was reached. Phases 3–5 constituted an iterative process of interviewing participants, discussing topics that emerged from interviews, and identifying new ones that needed more validation and deepening during upcoming interviews. Data collection was concluded when the research team agreed that sufficient data was collected on each theme and data saturation was achieved.

## Ethical considerations

This study was carried out in line with the Declaration of Helsinki of 1975 (as revised in 2013), the Dutch Medical Research Involving Human Subjects Act (WMO), and the General Data Protection Regulation (GDPR). It has been approved by the Medical Ethical Committee of the University Medical Centres (UMC) in Amsterdam, Netherlands (NL76469.029.21, registered under 2021.0158). All interviewees received an information letter and, after seven days of time to consider, gave their written consent by signing an informed consent. The information letter, the informed consent form and the procedure to obtain consent by signing an informed consent were approved by the Medical Ethical Committee of the Amsterdam UMC. Capacity (to consent) was assessed by qualified mental healthcare professionals. Patients received a 10-euro voucher for their participation.

## Results

### Execution of the intervention

At the start of the trial, 32 MHPs (including psychiatrists, nurses, social workers, psychologists and experts-by-experience) took part in the KISMET intervention and provided smoking cessation support to their patients over one year. At the end of the trial, a total of 206 sessions were carried out across all locations ranging between 16–20 sessions (average 18

sessions). Fourteen sessions were cancelled due to personnel shortages, holiday or sick leave of MHPs, or the absence of all patients.

### Attendance at group sessions

A total of 89 patients initially consented to participate in the KISMET intervention arm. However, 15 patients dropped out before the intervention started or did not complete a baseline assessment. Eventually, 74 patients began in the intervention arm. The average group size per FACT team was seven patients, ranging from five to 10 patients. Attendance at sessions in this selection of patients was on average 68% of the total sessions given.

### Drop-out of patients and MHPs

Across 11 FACT teams, eight (out of 32) MHPs dropped out of the RCT due to pregnancy leave, indefinite sick leave, starting a traineeship or changing jobs. Four out of these eight dropouts could be replaced with colleagues who were motivated to provide the KISMET intervention. The planned deployment of other dropouts could be compensated by the remaining colleagues involved in the study. In total, 52 patients (58%) had dropped out at the 12-month follow-up in the intervention teams. Increasing mental health problems or being admitted to a psychiatric ward were among the most named reasons for drop-out (28%). Other reasons included a loss of motivation (15%), no confidence in their own capacity to quit smoking (9%) or an unsuccessful quit attempt (6%).

### Characteristics of interviewees

The characteristics of all interviewed MHPs and patients are presented in Table 1.

### Interviews

Table 2 provides an overview of the main results yielded from the interviews.

### REACH

The recruitment of patients mainly relied on the MHPs responsible for carrying out the KISMET intervention. Most MHPs, however, perceived it as more effective if recruitment was shared by all FACT team members. Through the involvement of all team managers, MHPs noted that more colleagues were stimulated to approach patients from their caseload, having a positive impact on inclusion rates. According to MHPs, a closer relationship with a patient increased the patient's willingness to participate, as the MHP was able to motivate the patient on a more personal level. Hence, placing the recruitment process within the caseload of each FACT team member was the most effective and efficient way to optimise the inclusion of patients. This was only implemented in a few FACT teams.

*"If you know a client well, you have certain advantages. There is a greater chance that you can activate someone and create more motivation. Of course, I didn't know all the clients I approached, and I could feel that distance during recruitment." (MHP4)*

Low expectations regarding patients' motivation and success were described as influential as to how much effort was put into approaching and convincing these patients to participate. By this selective approach, some MHPs admitted that they might have missed on patients who potentially would have participated but were not approached.

Some patients named the long duration and high frequency of the group sessions as a reason for not wanting to participate. For other patients, the group setting was not preferred due to the risks of overstimulation. Overall, MHPs recommended preparing patients more gradually for participation through motivational enhancement techniques, thereby

**Table 1. Characteristics of interviewees.**

*A. Mental healthcare professionals*

| ID | Gender | Age | Profession | Total sessions given |
|---|---|---|---|---|
| MHP1 | Man | 50-59 | Nurse | 20 |
| MHP2 | Woman | 30-39 | Nurse | 20 |
| MHP3 | Woman | 60-69 | Nurse | 16 |
| MHP4 | Woman | 40-49 | Psychologist | 18 |
| MHP5 | Woman | 20-29 | Nurse | 19 |
| MHP6 | Woman | 50-59 | Clinical Nurse Specialist | 20 |
| MHP7 | Man | 40-49 | Social worker | 18 |
| MHP8 | Woman | 30-39 | Experience worker | 20 |
| MHP9 | Woman | 50-59 | Clinical Nurse Specialist | 20 |
| MHP10 | Woman | 50-59 | Clinical Nurse Specialist | 19 |

*B. Patients*

| ID | Gender | Age | Primary diagnosis | Smoking status at time of interview | Attendance of sessions |
|---|---|---|---|---|---|
| P1 | Man | 40-49 | Bipolar disorder | Past smoker | 90% |
| P2 | Woman | 40-49 | Psychotic disorder | Current smoker | 67% |
| P3 | Woman | 30-39 | Bipolar disorder | Current smoker | 55% |
| P4 | Man | 40-49 | Depressive disorder | Past smoker | 100% |
| P5 | Man | 40-49 | Schizophrenia | Past smoker | 95% |
| P6 | Woman | 30-39 | Eating disorder not otherwise specified | Current smoker | 63% |
| P7 | Woman | 50-59 | Schizophrenia | Current smoker | 87% |
| P8 | Man | 30-39 | Psychotic disorder | Current smoker | 25% |
| P9 | Man | 50-59 | Depressive disorder | Current smoker | 65% |
| P10 | Man | 40-49 | Schizophrenia | Current smoker | 85% |
| P11 | Man | 20-29 | Schizophrenia | Current smoker | 60% |
| P12 | Woman | 40-49 | Unspecified personality disorder | Current smoker | 63% |
| P13 | Woman | 40-49 | Bipolar disorder | Past smoker | 74% |
| P14 | Woman | 50-59 | Schizophrenia | Past smoker | 84% |
| P15 | Man | 40-49 | Bipolar disorder | Current smoker | 35% |
| P16 | Man | 20-29 | Schizophrenia | Past smoker | 50% |

decreasing the probability of early termination. According to MHPs, a patient's mental well-being should be taken into consideration as higher severity of symptoms often interfered with participation.

## EFFICACY

Interviewees evaluated the efficacy of all three intervention components: Cognitive-behavioural support group, peer support group and pharmacological treatment. In addition, the carbon monoxide (CO) measurement, one of the study assessments, was appraised.

**Cognitive behavioural support group.** Patients described psychoeducation about tobacco addiction, health risks of smoking and (addictive) ingredients in a cigarette to be an engaging introduction as it was considered both a confrontation and an 'eye opener'. Specifically, CBT-based exercises, such as registering which situations or emotions usually trigger the urge to smoke (more) yielded relevant insights. Patients acquired a deeper understanding of their underlying emotions that motivate smoking and could critically revise smoking as a routine response to negative emotions or stress. Likewise, MHPs corroborated the experiences shared by the patients and integrated them into setting new treatment goals.

 

**Table 2. Main results.**

| RE-AIM dimension | Mental healthcare professionals | Patients |
|---|---|---|
| Reach | • Recruitment best carried out by entire FACT team<br>• Patients should be prepared gradually for participation with motivational enhancement techniques<br>• Patient recruitment and subsequent retention was challenging<br>• Study attrition was a serious threat to implementation | • Group setting was an incentive to participate for some<br>• Burden of intervention seemed too high<br>• Dropout reasons included mental health problems, admission to psychiatric ward, and loss of motivation |
| Efficacy | • Group sessions and pharmacotherapy good combination<br>• CO monitoring is a feasible, user-friendly method<br>• Difficult to deliver intervention to diverse group of people due to differences in cognitive capacities | • CBT-based exercises considered helpful<br>• NRT highly effective to reduce withdrawal symptoms<br>• CO monitoring is motivating<br>• Group sessions and peer support group should be combined<br>• Smoother transition in changing frequency of group sessions is advised |
| Adoption | • Changing culture regarding smoking increased willingness to participate in KISMET intervention<br>• Shortage of financial and human resources was a major barrier | Not applicable |
| Implementation | • KISMET handbook offered excellent guidance and increased fidelity<br>• Peer support group was challenging due to limited availability of expert-by-experience<br>• Familiarisation with pharmacological treatment procedures is needed<br>• Effective teamwork is crucial for optimal implementation<br>• Shortage of staff and high turnover were experienced as obstacles | • Inconsistent presence of other patients was disrupting<br>• Regular attendance was difficult due to mental health status and fluctuating motivation<br>• Pharmacological treatment needs improved implementation from MHPs |
| Maintenance | • Intention to continue KISMET intervention<br>• Proper organisational embedding is needed | Not applicable |

*"I enjoyed it very much; it's a nice group of people. It was also great to work towards something positive together, towards a healthier lifestyle and its benefits. Not just talking about problems." (MHP2)*

Patients emphasized that addressing the interaction of smoking and (the severity of) their psychiatric symptoms was indispensable in the treatment of tobacco addiction. According to patients, more efforts should be devoted to the function of smoking for stress management, since experiencing distress was the most likely reason for relapse. At times, MHPs noted that it was challenging to deliver the sessions to a diverse group of people. Different levels of cognitive capacity and concentration lead to discrepancies in how much people could follow and engage in the group sessions. The experience of psychotic and, particularly, negative symptoms, as well as using antipsychotic medication, impeded group participation. MHPs described it as difficult to balance the amount of information and pace of conversation to make the group sessions stimulating and manageable for every participant.

Overall, patients experienced the group setting as enriching. The three months with weekly sessions was considered too short, however. Primarily because the process of preparing for a quit attempt took more time than expected. Most patients made a quit attempt towards the end of the three-month intensive phase. Consequently, decreasing the frequency of sessions at this point was not helpful as some people still needed weekly support. The preferred frequency of meetings also depended on patients' current mental health status. For example, starting a smoking cessation trajectory parallel to other intensive psychological treatments, such as Eye Movement Desensitization and Reprocessing (EMDR), weekly meetings were too frequent.

**Peer support group.** Both patients and MHPs described the concept of peer support as beneficial. Working with a (former) patient or MHP who has successfully quit smoking was considered helpful. Experts-by-experience who were interviewed recognised appreciation from patients upon sharing their experiences with nicotine withdrawal, relapse, health benefits after smoking cessation and stress management. Furthermore, sharing information and experiences regarding

the use of medication for smoking cessation was valued by patients; it helped to diminish doubts and created more optimistic expectations towards medication use and its effectiveness. In the original implementation plan, peer support group meetings were scheduled weekly on the same day as the group sessions. However, patients found it exhausting to attend two consecutive hours of group sessions. Patients and MHPs suggested combining both groups into one session.

**Pharmacological treatment.** Twenty-one patients (28.4%) participating in the KISMET intervention indicated that they received pharmacological treatment for smoking cessation at 12-month follow-up. Thirteen patients (17.5%) used NRT (nicotine patches, melt tablets and/or chewing gum). Eight patients (10.8%) used other pharmacological support, i.e., Cytisine, Bupropion, or Nortriptyline, whether in combination with NRT. Overall, patients reported minimal side effects and regarded NRT as highly effective in coping with withdrawal symptoms and severe cravings. By alleviating these symptoms, patients felt they were able to focus more on the psychological aspects of their tobacco addiction and cessation.

*"It kind of gave me the feeling that half of the battle was already won, […] because the patches gave me some kind of peace, physically." (P6)*

Since Bupropion and Nortriptyline are antidepressants, psychiatrists did not prescribe them if other antidepressants were already prescribed or were generally advising against the use of medication for smoking cessation due to the potential side effects and interactions with antipsychotic medications. Complicated prescription procedures and the unavailability of psychiatrists hindered the accessibility and effective use of smoking cessation medication. Patients from some FACT teams complained about long waiting times for a consultation appointment and delays in the delivery of medication. These complications also influenced patients' planning of their quit attempt, as well as their overall motivation and readiness to quit.

*"Most of us were supposed to see a psychiatrist before the intervention started. Well, that wasn't the case, and some people only got a consultation for medication after six months. [...] So it was a long wait and that eats away at your motivation." (P9)*

**Carbon monoxide (CO) monitoring.** Even though not a part of the original intervention protocol, but part of the study assessments, CO monitoring was mentioned as a helpful tool by patients and MHPs.

*"The CO measurement was always a confrontation with how much they smoked and what effect it has on their body. I think that it is a good and helpful method." (MHP2)*

During the group sessions, patients were motivated by the results of the CO monitoring. When their CO levels were low, indicating that they had smoked few or no cigarettes shortly before the measurement, patients felt encouraged by seeing the immediate results of their efforts. This boosted their motivation to remain abstinent or to further reduce their daily cigarette intake. On the other hand, when their CO levels were high, patients saw the monitoring as a confrontation.

*"Sometimes my CO level was very low, which I was glad about. Other times it was that of an extremely heavy smoker. So, it was very confrontational, but good to do it and get back on track." (P2)*

## ADOPTION

The Dutch national agreement on smoke-free mental healthcare institutions (by 2025) and – as a part of this – policies prohibiting smoking on institutions' territory played a significant role in the decision to implement the KISMET intervention.

The participating MHPs expressed intentions to act in line with these national policies but observed the lack of practical tools and skills. For them, the KISMET intervention was a welcomed programme. Some MHPs expressed frustration with standard care for smoking cessation since current guidelines are often neglected in practice. Changing existing cultures and practices regarding smoking is most likely to be a slow and thus a lengthy process, considering smoking has been not only accepted and normalised for many decades in mental healthcare but also instrumentalised for stress management in patients. These shortcomings of appropriate support stimulated professionals to participate.

## IMPLEMENTATION

The following paragraph discusses facilitators and barriers to implementation from the patient level to the organisational level.

**Patient level.** Patients' adherence to the group sessions was a crucial factor for a smooth implementation. In all FACT teams, patients noted their inconsistent attendance as well as that of others. Various reasons were given for inconsistent attendance, such as increased severity of psychiatric symptoms, low motivation, or other commitments such as psychotherapy. Patients who attended regularly described feeling left alone by other patients in the group who dropped out or were often absent. Accumulatively, the inconsistency in presence disrupted the overall flow of implementing the KISMET intervention as sessions were occasionally cancelled or held short, due to the absence of patients.

**MHP level.** Twenty-six MHPs (out of 32) attended the KISMET training. Most MHPs expressed that they found the training a good preparation for implementing the intervention. There was variation in prior knowledge and skills regarding smoking cessation support, CBT and MI techniques. MHPs indicated that their preexisting knowledge and experience heightened their confidence in their ability to deliver the intervention, potentially leading to better implementation.

Professionals agreed that the KISMET handbook offered excellent guidance and instructions through detailed descriptions of the sessions' content. MHPs appreciated the consistent structure of the handbook and noted that it elevated their confidence regarding the delivery of the intervention. This significantly contributed to treatment fidelity. Occasionally, professionals deviated from the protocol regarding the duration of the sessions or spread the content of one session over multiple sessions. Adjustments were made in response to patients' needs, preferences and capacities. We observed nine group sessions in nine (out of 11) intervention teams. All teams adhered to the content of the session as protocolised in the handbook. The structure of the sessions as proposed in the handbook (1. check-in; 2. group discussion; 3. goal setting for the following weeks) was less complied with, as four FACT teams did not have a check-in in the beginning and did not finish the session with goal setting for the following weeks. Regarding peer support, most FACT teams could not implement regular meetings due to the unavailability of staff.

Professionals' beliefs regarding patients' autonomy were decisive in how they experienced their patients' participation in the intervention. MHPs who attributed high autonomy to their patients reported less frustration about a stagnating process regarding smoking cessation or unsuccessful quit attempts. More specifically, these MHPs portrayed the process of quitting entirely owned by the patients themselves, while meeting the patients' need for support.

*"We try to encourage patients to look at the disadvantages of smoking and the advantages of quitting. But ultimately everyone remains responsible for their process. You must slow yourself down so that you don't fall into the trap of pushing and pushing." (MHP1)*

This can benefit implementation as professionals hold more positive attitudes regarding the KISMET intervention. Some MHPs admitted to having pessimistic expectations regarding the chances to quit smoking of some individuals in this patient group.

**Relationship MHP – patient.** Mutual trust to commit to the group sessions and the common goal of working towards smoking cessation was perceived as the fundament for a strong collaboration, according to MHPs.

Patients' lack of compliance with the group sessions weakened the trust of MHPs in the chances for successful quitting. According to patients, this sometimes expressed itself in a vicious cycle of pessimistic expectations by MHPs and patients who perceived this lack of confidence and therefore complied even less. For patients, compliance by MHPs critically contributed to the feeling of being taken seriously. Extra efforts by the MHPs to do reminder calls or messages before the sessions were appreciated by patients. Irregularity or cancellations of group sessions demotivated patients and harmed the therapeutic alliance and was therefore an obstacle to implementation.

*"At a certain point, I had the feeling that X and Y* [MHPs] *were not that motivated because group sessions were rescheduled or cancelled very often. And then I started to feel that in the other clients, too. […] I don't think that anyone felt like they were being taken seriously anymore." (P6)*

**FACT team level.** Patients and MHPs from FACT teams in which there was high regularity and consistency in the delivery of the intervention mainly assigned this to effective teamwork and communication within the entire FACT team. MHPs evaluated the implementation more positively when they thought that the collaboration with their colleagues was effective. Support from colleagues not directly involved in the implementation of KISMET included, for instance, following up with patients from their caseload who participated in the KISMET intervention and reporting back to their colleagues delivering it. However, it was also noted that communication with other team members was often difficult due to the way how FACT teams work, including frequent house visitations, limiting contact on location with colleagues. Moreover, a shortage in clinical staff and turnover were reported by the majority of MHPs making teamwork more challenging.

The shortage in staff and lack of collaboration was particularly noted among psychiatrists. Pharmacological treatment for smoking cessation depended on the cooperation between psychiatrists/ clinical nurse specialists (CNS) and MHPs delivering the group sessions. Consequently, disruptions in communication resulted in consultation appointments later than patients requested and little or no monitoring. In addition, many psychiatrists and CNSs had little experience with the prescription procedures, especially for NRT. In FACT teams that reported the least complications around implementing pharmacological treatment, consultation appointments were scheduled by default for every patient at the beginning of the intervention.

**Organisational level.** Lastly, MHPs critically evaluated the implementation on an organisational level. Unanimously, more embedding is necessary to facilitate the implementation of the KISMET intervention. Professionals suggested that managers should take a more active role in the planning and organisation, for example, by ensuring that FACT teams are staffed sufficiently to carry the extra workload and compensate where needed. Moreover, to make it a higher priority among all FACT team professionals and to stimulate their proactivity with patients regarding smoking cessation.

### MAINTENANCE

MHPs indicated the willingness and motivation to continue a smoking cessation group within their FACT team and organisation. They also clearly stated that for better implementation of a smoking cessation programme, proper embedding within the organisation is essential. Offering the programme to more than one FACT team across one organisation might facilitate the engagement of patients. Despite the optimistic intentions to maintain smoking cessation support for their patients, some MHPs also expressed concerns about the willingness on a managerial level due to financial shortcomings.

## Discussion

This study documents the outcomes of a process evaluation exploring the perceptions of MHPs and patients of the implementation of the KISMET intervention.

In the beginning, FACT teams experienced difficulties in recruiting patients due to their unwillingness to participate. The reasons patients gave for this included the high burden of the intervention. Reaching this specific patient group is conflicted: the aim to reduce the burden for patients directly opposes that appropriate tobacco addiction treatment entails intensive and long-term support to increase the chances of treatment success. While a shorter intervention period might facilitate recruitment it might compromise the intervention's effectiveness. However, the desired duration of the intervention can differ between individuals and future research could investigate the effectiveness of shorter interventions in this patient group.

Overall, reports on CBT-based group sessions are in line with the evidence for the effectiveness of CBT and motivational techniques for tobacco addiction treatment [12,16,29]. Results regarding peer support also align with recent research on the effectiveness of peer support for smoking cessation in people with severe mental illness [30,31]. Unfortunately, in almost all FACT teams, the peer support group could not be delivered consistently due to restricted availability or the unexpected absence of an expert-by-experience because of relapse (e.g., psychosis or smoking). Peer support as part of an intervention is expectedly more vulnerable to inconsistent delivery due to the higher vulnerability of experts-by-experience themselves. At the same time, this vulnerability probably contributes to mutual understanding in the relationship between the expert-by-experience and patient, making this collaboration powerful. In some FACT teams, MHPs who quit smoking would deliver peer support sessions. Still, regularity in session could not be secured due to the MHP's lack of availability. Despite the suboptimal implementation, the testimonies of patients and professionals favour the integration of peer support in smoking cessation interventions, confirming what was established in our Delphi study [22].

Pharmacological treatment was underutilised with 57% of patients reporting that they had used pharmacological support. This is in line with recent research that showed a decline in prescriptions of medication for tobacco addiction in people with SMI [32]. One underlying reason that emerged from our study was the relatively new procedures for prescription. In 2022 a new reimbursement system with new regulations was introduced in the Dutch mental healthcare sector which increased the risk for systemic errors during prescription and reimbursement procedures [33]. Overall, patients positively appraised pharmacological treatment while also pointing out that psychiatrists and CNSs need to familiarise themselves more deeply with prescription procedures. Additionally, professionals should gather more knowledge about medication to support smoking cessation and potential interactions with other medications.

Generally, CO monitoring is used in research to biochemically verify smoking status with the outcome in parts per million (ppm) [34,35]. Only very few studies use more frequent CO monitoring as a method to reduce or quit smoking [36]. In the KISMET trial, CO monitoring was planned solely as part of data collection but emerged as a valuable method for patients during their smoking cessation process. Both patients and professionals implied that the CO monitor served as an insightful and motivational tool and suggested its systematic integration into smoking cessation programs. These findings are in line with recent studies on the beneficial effect of self-monitoring on substance use [37], further advocating the use of CO monitoring to support smoking cessation.

Prominently, all MHPs underlined that a lack of organisational embedding of the KISMET intervention was the biggest barrier to implementation. An optimal embedding within mental healthcare organisations includes the systematic planning of implementation of an intervention on the managerial and healthcare provider levels and the allocation of financial and human resources. The current lack of such embedding hinders the innovative advancement and integration of interventions, such as KISMET, in FACT teams. Moreover, the chances that FACT teams commit to implementing future interventions might be decreased due to the anticipated lack of organisational embedding and managerial support. Team managers should set up a structured implementation plan to determine roles and assigning responsibilities as well as monitoring the implementation process and holding regular meetings to ensure efficient coordination among MHPs.

## Strengths and limitations

There are several strengths to this study. In total, we conducted twenty-six interviews, of which sixteen were with patients. Firstly, the number of interviews contributed significantly to the comprehensive evaluation of the KISMET intervention from

both patients' and professionals' perspectives. The ratio between interviews with patients and professionals strengthens one of the main aims of this research: to purposefully capture patients' experiences to further develop smoking cessation interventions in line with patients' needs and preferences. Secondly, we created the interview guides based on the RE-AIM framework that ensured accurate data collection to assess all significant dimensions of the implementation process. Thirdly, the interviews were conducted within the active study period to increase the reliability of the testimonies due to their proximity in time.

There are a few limitations: since the interviews were with the researchers of the KISMET trial, MHPs might have answered in a socially desirable way [38]. To minimise potential response bias, we emphasised that all feedback is valuable, including challenges they encountered. We attempted to normalise problems with implementation and presented interviewees with positive and negative examples from other participating centres. Another limitation is the mode of the interviews with MHPs. All interviews with patients were in person. The interviews with professionals, however, were all conducted online via Zoom. While online interviews have a practical advantage, especially with FACT teams in various geographical regions, online interviewing might hinder the rapport between the interviewer and the interviewee. This can come at a cost to the depth of the conversation and to what extent MHPs were willing to open up about delicate topics [39]. To facilitate the latter, we carefully assigned the professionals to the interviewers (MK and LJ) based on who worked with whom most intensively throughout the research. Lastly, our study did not cover the perspective of FACT team managers or board members of mental healthcare institutions to shed light on the suboptimal implementation on an organisational level.

### Implications for research

To facilitate inclusion and reduce study attrition, recruitment strategies should include a motivational phase and foster a closer therapeutic relationship as it benefits recruitment [40]. The high drop-out rates in our study are in line with the findings from a recent systematic review [41]. Motivational enhancement techniques could reduce dropout rates by engaging patients to continue with the intervention. Additionally, normalising fluctuations in motivation can reinforce the acceptance of demotivation in patients, thereby making it part of the process rather than a reason to stop. According to MHPs' perceptions, most patients who dropped out showed a higher degree of cognitive inflexibility regarding quitting smoking. Tackling this black-and-white thinking style, for instance by un-labelling relapse as a failure, can thereby help to reduce study attrition.

To gain more direct insights into the process of implementation researchers should do more on-site observations as they are opportunities to understand the challenges and possibilities in real-life clinical settings, in which the research is carried out. Finally, to gain first-hand insights on managerial, financial and organisational barriers and facilitators to implementation it is recommended to conduct further interviews with managers and board members.

### Conclusions

The results of this study can guide addressing barriers and embracing facilitators for the future implementation of smoking cessation interventions in ambulatory mental healthcare settings.

### Supporting information

**S1 File. Supporting information file.**
(DOCX)

### Acknowledgments

The authors would like to thank all FACT teams and patients for their participation from GGZ Delfland, GGZ inGeest, Antes, Parnassia Den Haag, GGZ Oost Brabant, GGZ Drenthe, GGNet, GGZ Noord-Holland-Noord, GGz Centraal, Mondriaan, and Pro Persona.

## Author contributions

**Conceptualization:** Müge H. Küçükaksu, Trynke Hoekstra, Marcel C. Adriaanse, Berno van Meijel.

**Data curation:** Müge H. Küçükaksu, Lola Jansen, Marcel C. Adriaanse, Berno van Meijel.

**Formal analysis:** Müge H. Küçükaksu, Lola Jansen.

**Funding acquisition:** Marcel C. Adriaanse, Berno van Meijel.

**Methodology:** Trynke Hoekstra, Berno van Meijel.

**Project administration:** Müge H. Küçükaksu, Lola Jansen, Sanne Helmig.

**Supervision:** Trynke Hoekstra, Marcel C. Adriaanse, Berno van Meijel.

**Writing – original draft:** Müge H. Küçükaksu.

**Writing – review & editing:** Müge H. Küçükaksu, Trynke Hoekstra, Marcel C. Adriaanse, Berno van Meijel.

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
