## [Decision Letter · Decision Letter 0]

14 Feb 2025

PONE-D-24-56086Implementation of a smoking cessation intervention for people with severe mental illness in ambulatory mental healthcare (KISMET): a process evaluationPLOS ONE

Dear Dr. Küçükaksu,

Thank you for submitting your manuscript to PLOS ONE. After careful consideration, we feel that it has merit but does not fully meet PLOS ONE’s publication criteria as it currently stands. Therefore, we invite you to submit a revised version of the manuscript that addresses the points raised during the review process.

**ACADEMIC EDITOR: **This is very important study that contributes to efforts in reducing smoking among mental healthcare professionals and patients. The findings are critical not only for people with severe mental illness but also can be adapted to general population. The quality of the study and presentation is near satisfactory. The authors should please address the following observations as well as the comments provided by Reviewer 1 (see below). **Funding statement** :

Please add agreement/contract number. Also indicate whether the funder played any role in conducting the study.

**Methods:**

MHPs and patient selection and recruitment:

The selection of MHPs and patients is not clearly defined. Please provide inclusion and exclusion criteria for both MHPs and patients selection. This is in addition to the variations provided.

Data analysis:

Please clearly explain the approach used in coding the transcripts – inductive/deductive or both. Was coding framework developed? E.t.c

We look forward to receiving your revised manuscript.

Kind regards,

Ibrahim Jahun, MD, MSC, PhD

Academic Editor

PLOS ONE

Journal Requirements:

2. Please describe in your methods section how capacity to provide consent was determined for the participants in this study. Please also state whether your ethics committee or IRB approved this consent procedure. If you did not assess capacity to consent please briefly outline why this was not necessary in this case.”

“This four-year research project is funded by the foundation Stichting tot Steun VCVGZ in the Netherlands.”

5. In this instance it seems there may be acceptable restrictions in place that prevent the public sharing of your minimal data. However, in line with our goal of ensuring long-term data availability to all interested researchers, PLOS’ Data Policy states that authors cannot be the sole named individuals responsible for ensuring data access (http://journals.plos.org/plosone/s/data-availability#loc-acceptable-data-sharing-methods).

8. Please review your reference list to ensure that it is complete and correct. If you have cited papers that have been retracted, please include the rationale for doing so in the manuscript text or remove these references and replace them with relevant current references. Any changes to the reference list should be mentioned in the rebuttal letter that accompanies your revised manuscript. If you need to cite a retracted article, indicate the article’s retracted status in the References list and also include a citation and full reference for the retraction notice.

**Reviewers' comments:**

Reviewer's Responses to Questions

Comments to the Author

1. Is the manuscript technically sound, and do the data support the conclusions?

Reviewer #1: Yes

Reviewer #2: Yes

2. Has the statistical analysis been performed appropriately and rigorously? 

Reviewer #1: I Don't Know

Reviewer #2: Yes

3. Have the authors made all data underlying the findings in their manuscript fully available?

Reviewer #1: No

Reviewer #2: Yes

4. Is the manuscript presented in an intelligible fashion and written in standard English?

Reviewer #1: Yes

Reviewer #2: Yes

5. Review Comments to the Author

Reviewer #1: The current project aimed to explore the perceptions of patients and mental health providers on the implementation of a smoking cessation intervention for individuals with severe mental illness. The mixed methods design provides critical insights for implementing smoking cessation interventions with difficult to treat populations. Overall, this very strong and sound study provides important information including the successes and difficulties of the KISMET program. A couple of minor notes: P3 L66-68 states that research on the combination of CBT, MI and NRT for smoking cessation is lacking; however several studies have begun looking at the effectiveness of these combined approaches (albeit in different populations, different timescales, and/or selective components of CBT, MI, pharmacotherapy) (see Brett et al., 2021; okuyemi et al., 2013 Wittchen et al., 2011; Zhou et al., 2023. Additionally, P20 L453-454 stated that shorter intervention periods may compromise the intervention effects. Brett et al (2021) found encouraging results for up to 6months after a single-session brief intervention. Although different, this suggests that shorter interventions may merit consideration and investigation, given the high drop out in the current project. Finally, L480-481 discuses the impact of CO monitoring. The paper may benefit from research on other substance use on how self-monitoring can lead to reductions in substance use (e.g., Gass et al., 2021).

Brett, E. I., Chavarria, J., Liu, M., Hedeker, D., & King, A. C. (2021). Effects of a brief motivational smoking intervention in non-treatment seeking disadvantaged Black smokers. Journal of Consulting and Clinical Psychology, 89(4), 241.

Gass, J. C., Funderburk, J. S., Shepardson, R., Kosiba, J. D., Rodriguez, L., & Maisto, S. A. (2021). The use and impact of self-monitoring on substance use outcomes: A descriptive systematic review. Substance Abuse, 42(4), 512-526.

Okuyemi, K. S., Goldade, K., Whembolua, G. L., Thomas, J. L., Eischen, S., Sewali, B., ... & Des Jarlais, D. (2013). Motivational interviewing to enhance nicotine patch treatment for smoking cessation among homeless smokers: a randomized controlled trial. Addiction, 108(6), 1136-1144.

Wittchen, H. U., Hoch, E., Klotsche, J., & Muehlig, S. (2011). Smoking cessation in primary care–a randomized controlled trial of bupropione, nicotine replacements, CBT and a minimal intervention. International journal of methods in psychiatric research, 20(1), 28-39.

Zhou, L., Guo, K., Deng, X., Shang, X., E, F., Xu, M., ... & Li, X. (2023). Effects of different combined behavioral and pharmacological interventions on smoking cessation: a network meta-analysis of 103 randomized controlled trials. Journal of Public Health, 1-11.

Reviewer #2: This insightful study examines the implementation of the KISMET smoking cessation program in Dutch mental healthcare. Recruitment and retention posed significant challenges, with a high drop-out rate (58%) emphasizing the need for stronger motivational strategies. Patients found group and peer support valuable, though inconsistent attendance affected engagement. While pharmacological treatment was beneficial, better facilitation was needed, and staff shortages, along with limited collaboration, hindered implementation. The findings highlight the crucial role of teamwork, structured support, and effective supervision in strengthening smoking cessation programs for individuals with severe mental illness.

6. PLOS authors have the option to publish the peer review history of their article (what does this mean? ). If published, this will include your full peer review and any attached files.

**Do you want your identity to be public for this peer review?** For information about this choice, including consent withdrawal, please see our Privacy Policy .

Reviewer #1: No

Reviewer #2: No

---

## [Author Response · Author response to Decision Letter 1]

3 Mar 2025

Dear Editor and Reviewers,

We would like to thank your for your positive review and suggestions to improve our manuscript. In the 'Response to Reviewers' file we have addressed each of your comments.

Thank you again and best regards,

Müge Küçükaksu

---

## [Decision Letter · Decision Letter 1]

17 Mar 2025

Implementation of a smoking cessation intervention for people with severe mental illness in ambulatory mental healthcare (KISMET): a process evaluation

PONE-D-24-56086R1

Dear Dr. Handan Küçükaksu,

We’re pleased to inform you that your manuscript has been judged scientifically suitable for publication and will be formally accepted for publication once it meets all outstanding technical requirements.

Kind regards,

Ibrahim Jahun, MD, MSC, PhD

Academic Editor

PLOS ONE

Additional Editor Comments (optional):

Reviewers' comments:

Reviewer's Responses to Questions

**Comments to the Author**

1. If the authors have adequately addressed your comments raised in a previous round of review and you feel that this manuscript is now acceptable for publication, you may indicate that here to bypass the “Comments to the Author” section, enter your conflict of interest statement in the “Confidential to Editor” section, and submit your "Accept" recommendation.

Reviewer #1: All comments have been addressed

2. Is the manuscript technically sound, and do the data support the conclusions?

Reviewer #1: Yes

3. Has the statistical analysis been performed appropriately and rigorously? 

Reviewer #1: Yes

4. Have the authors made all data underlying the findings in their manuscript fully available?

Reviewer #1: Yes

5. Is the manuscript presented in an intelligible fashion and written in standard English?

Reviewer #1: Yes

6. Review Comments to the Author

Reviewer #1: The authors have addressed all of my concerns. This manuscript provides important context on the implementation of smoking cessation treatment among difficult to treat populations.

7. PLOS authors have the option to publish the peer review history of their article (what does this mean? ). If published, this will include your full peer review and any attached files.

**Do you want your identity to be public for this peer review?** For information about this choice, including consent withdrawal, please see our Privacy Policy .

Reviewer #1: No

---

## [Editor Report · Acceptance letter]

PONE-D-24-56086R1

PLOS ONE

Dear Dr. Küçükaksu,

I'm pleased to inform you that your manuscript has been deemed suitable for publication in PLOS ONE. Congratulations! Your manuscript is now being handed over to our production team.

Kind regards,

on behalf of

Dr. Ibrahim Jahun

Academic Editor

PLOS ONE